# Isolated Growth Hormone Deficiency and Idiopathic Short Stature: Comparative Efficiency after Growth Hormone Treatment up to Adult Height

**DOI:** 10.3390/jcm10214988

**Published:** 2021-10-27

**Authors:** Ana-Belen Ariza-Jimenez, Isabel Leiva Gea, Maria Jose Martinez-Aedo Ollero, Juan Pedro Lopez-Siguero

**Affiliations:** 1Pediatric Endocrinology, Hospital Regional Universitario de Málaga, 29010 Málaga, Spain; mj.martinezaedo.sspa@juntadeandalucia.es (M.J.M.-A.O.); lopez.siguero@gmail.com (J.P.L.-S.); 2Pediatric Research, Fundación Pública Andaluza Para la Investigación de Málaga en Biomedicina y Salud, 29010 Málaga, Spain; 3Pediatric Research, Instituto de Investigación Biomédica de Málaga, 29010 Málaga, Spain

**Keywords:** growth hormone treatment, growth hormone deficiency, idiopathic short stature, final height

## Abstract

Introduction: Treatment with growth hormone (GH) is not approved for idiopathic short stature (ISS) in Europe. Objectives: To compare the growth of children treated with isolated GH deficiency (IGHD) vs. ISS-treated and untreated children. Methods: A retrospective descriptive study of patients treated in the last 14 years for IGHD (Group A), in comparison with ISS-treated (Group B) and untreated (Group C) subjects. Results: Group A had 67 males, who showed a height gain of 1.24 SD. Group B had 30 boys, who showed a height gain of 1.47 SD. Group C had 42 boys, who showed an improvement of 0.37 SD. The final heights were −1.52 SD, −1.31 SD, and −2.03 SD, respectively. Group A and C did not reach their target heights (with differences of 0.27 SD and 0.59 SD, respectively). Group B surpassed their target height by 0.29 SD. Conclusions: The final heights of the IGHD and treated ISS are similar. Treated groups were taller than untreated groups.

## 1. Introduction and Background

Since normal growth hormone (GH) secretion is pulsatile, with very low levels during the day, and occurs more frequently during sleep, with six or eight spontaneous peaks, clinicians are not able to rely on baseline GH levels. Thus, they stimulated the pituitary somatotrophs, either pharmacologically or physiologically, in order to observe the response in terms of GH secretion [1]. The reproducibility of GH stimulation tests varies according to the stimuli that is used [2]. Since normal children may not respond to a specific stimulus test (high rate of false positives), the absence of a response must be confirmed with a second test using a different stimulus [1,3].

There are several studies that criticize the pharmacological tests used for the diagnosis of isolated GH deficiency [4,5,6]. In fact, these tests have significant issues as they do not mimic a normal secretory pattern of GH, and there is an arbitrary definition of normal levels, a low specificity (up to 60% of normal children have a deficient GH response), and poor variability in outcomes, depending on multiple variables, such as the type of stimulus, the type of trial, psychosocial factors, age, body mass index (BMI), pubertal stage, and the use of sex steroid priming [1,3]. These tests are also expensive, uncomfortable, and may sometimes involve an element of risk to the patient [1,7,8].

Accordingly, there is disagreement as to whether patients with Growth Hormone Deficiency (GHD) respond better to GH than patients with Idiopathic Short Stature (ISS), with some of the difference likely being due to how strictly GHD is defined. In fact, patients with permanent GHD yield better results than those with transitory GHD, just as patients with multiple pituitary deficiencies show better results than those with isolated growth hormone deficiency [9,10].

Therefore, due to the controversy over functional tests of GH and the effectiveness of GH treatment for short stature, we proposed a comparison of groups with positive and negative results in GH stimulation testing in order to evaluate the effectiveness of GH therapy in these patients.

Hypothesis: There are no significant differences between the adult heights of patients with ISS and those with isolated GHD when both are treated with GH.

Aims: 1. To evaluate the differences between the adult heights of patients with idiopathic short stature and isolated GH deficiency, both treated with GH. 2. To compare treated isolated GH deficiency and treated idiopathic short stature patients with a historic non-treated idiopathic short stature group.

## 2. Materials and Methods

We performed a longitudinal retrospective descriptive study on a cohort of male children who were diagnosed—via the application of two stimulation tests—with isolated GH deficiency, between 2 and 14 years of age, and were treated with GH until adult height was reached (Group A).

This cohort was compared with two previous studies in our hospital consisting of male children, diagnosed with idiopathic short stature at ages between 2 and 14 years, who were either treated with GH (Group B) or not treated with GH (Group C) [11,12].

All measurements were performed by trained and skilled health professionals. To avoid measurement errors, they were repeated three times. Additionally, our patients were measured in underwear and without shoes. Weight was measured using a manual Seca brand bascule with an accuracy of 0.1 kg. Height was measured using a Holtain Stadiometer with 0.1 cm precision, as calibrated daily.

Stimulation tests were performed with exercise, and clonidine was used for the diagnosis of childhood GHD or ISS, with a cutoff of 7 ng/mL [2,13]. Stimulation testing was performed on two different days. For exercise stimulation testing, we determined GH levels under basal conditions and after 20–40 min of moderate intensity exercise such as running, with a final heart rate of 120 beats per minute. Stimulation with clonidine consisted of administrating 0.15 mg/m^2^ of oral clonidine, and determining GH levels at baseline, 30, 60 and 90 min. Boys under 3 years old were stimulated by glucagon as a testing method. We used insulin stimulation tests for reevaluation, with a cutoff of 5.6 ng/mL, according to current references [13]. It was performed with 0.1 U/kg of intravenous insulin, and GH levels were determined at 0, 15, 30, 45, 60, 75, 90 and 120 min. During the insulin test, the glucose level had to decrease by at least 50% of the initial value, or reach less than 40 mg/dL, in order to give validity to the test. Symptomatic hypoglycemia was treated with 2 mL/kg of intravenous glucose at 10% concentration. 

Furthermore, in the GHD group, we used somatotropin that satisfied the IRP IS 98/574 standard, and the 22k rhGH isoform was used to provide harmonized GH assays, which used a cut off value of 7 ng/mL. For the ISS groups, the IS 88/624 standard was used. Priming of sex steroids was not conducted for GH stimulation testing. Analytical results, both GH and IGF1, were processed in Malaga’s Children Hospital laboratory by qualified professionals using chemiluminescent immunometric assays (IMMULITE 2000). Due to the evolution of laboratory techniques throughout the 14 years of study, there may have been differences in the methodologies of hormone assays.

All patients were treated with biosynthetic growth hormone, which was applied at a median dose of 0.028 mg/kg/day (range: 0.02–0.036 mg/kg/day) for a mean of 5 years of treatment from prepubertal patients to adult patients. GH posology was chosen and adjusted over time according to weight and IGF1.

The cohort was monitored from diagnosis until adult height, while data on a number of variables were collected, including: age at start of treatment; standard deviation scores for weight; height; target height and body mass index at baseline, and at one, two, three, and four years of treatment; bone age/chronological age (BA/CA); predicted final height according to the Bayley–Pinneau Method; IGF1 at baseline and after one year of treatment; age at pubertal onset; height at pubertal onset; total growth during puberty; and adult height.

The target height was calculated as (mother’s height + father’s height)/2 ± 6.5 cm.

Data were adjusted automatically with the AUXOTEC^®^ software using standard deviations according to the tables provided in the Spanish transverse growth study [14]. IGF1 was adjusted according to age and sex using IMMULITE software.

### 2.1. Inclusion Criteria for Growth Hormone Deficiency


Male children that were younger than 14 years old who:◦Had heights of more than 2 SD below the mean for age and sex;◦Had growth velocities of less than −1 SD for chronological age in the last year, or of less than −1.5 SD over the past 2 years, according to the low-growth-rate definition;◦Had received two GH stimulation tests (exercise and clonidine) with GH < 7 ng/mL and delayed bone maturation of greater than or equal to 1 year;◦Showed an absence of associated pathology;◦Remained prepubertal at least during the 1st year of GH therapy.


### 2.2. Inclusion Criteria for Idiopathic Short Stature:


Male children that were younger than 14 years old who:◦Had heights of more than 2 SD below the mean for age and sex;◦Remained prepubertal at least during the 1st year of GH therapy;◦Had growth velocities of less than −1 SD for chronological age in the last year, or of less than −1.5 SD over the past 2 years, according to the low-growth-rate definition;◦Had a GH > 7 ng/mL after stimulation test;◦Had a birth weight > 2500 g.


### 2.3. Exclusion Criteria

Children with short stature due to other causes (hypothyroidism, hypercortisolism, chronic systemic diseases, dysmorphic syndromes, skeletal disorders, etc.);Patients with other associated hormone deficiencies, significant anatomical abnormalities, tumor, hypothalamic-pituitary disease, or genetic alterations;

Individuals who had ingested growth-promoting substances.

### 2.4. Ethical Issues

Participation in this study did not confer additional risk to participants since it was carried out according to the routine clinical practice protocols of the Pediatric Endocrinology department at Malaga Children’s Hospital, which are based on the recommendations of the Andalusian Committee for Growth, through measures, tests and prescriptions of growth hormone treatment according to these criteria, which are listed within our inclusion criteria. Furthermore, informed consent was received from parents.

### 2.5. Statistical Analysis

For descriptive and inferential statistical analysis, conducted through the use of bivariate and multivariate formulas, the SPSS Statistics statistical package, version 22, was used. The inferential normality test was performed using the Shapiro–Wilk test to check if there was a normal distribution of the data series, and the Levene test was used to determine homoscedasticity. In addition, the mean and standard deviation were used to express the centrality and variability of the sample, respectively.

In normal samples, the Pearson correlation was used in order to relate two quantitative variables. Logistic regression was used for investigation of the effect of several parameters on dichotomous variables. In addition, for qualitative comparisons with more than two categories, and for abnormal samples, the Kruskal–Wallis test was used. On the other hand, for abnormal samples with dichotomous categories, the Mann–Whitney U test was used.

Finally, multiple regression was used for all variables with respect to adult height

## 3. Results

A total of 67 Spanish male children with GHD were studied in Group A. The mean age at the onset of treatment was taken to be 9.9 ± 2.53 years old. Subjects within this group reached an adult height of −1.53 SD, with a total mean gain of 1.26 SD. The mean BMI in these patients was −0.42 SD (−2.04 to 2.56 SD) at onset. Results for all variables are shown in Table 1.

To compare the GHD patients with the ISS patients, we used data from our center published by López-Siguero et al. [11]. They studied the spontaneous growth to adult height of 42 male children with ISS (Group C), who presented, at onset, an average age of 10.8 ± 2.2 years. Of them, 24 had familial short stature and 17 had constitutional delays in growth and development. These male children showed improvements of 0.37 SD in relation to their initial heights, which entailed gains of 2.4 cm to 4.5 cm, even though they did not reach their target heights (difference of 0.59 SD or 3.5 cm). Age, initial height, and predicted height were the main predictors of adult height in these children [11], whereas in boys of Group A, the main predictors were height at the fourth year of treatment, age at pubertal onset, and the total increase in height during puberty [15]; therefore, both groups did not present any common variable in the prediction of adult height. The improvement in height in Group C was lower than the improvement found in Group A [11]. (Figure 1).

López-Siguero et al. also studied the evolution of height up until the reaching of adult height in 30 male children with ISS who were treated with GH at a dose of 0.028 mg/kg/day (0.07 SD) (Group B) [12]. They gained, on average, a height of 1.47 SD (8.8 cm) before adult height was reached. Thus, they concluded that GH treatment significantly increased the adult height of children with ISS. They found that target height and predicted height were independent predictors of adult height, with a regression coefficient of 0.38. In addition, they observed a difference between adult height and target height at a value of 0.29 SD (1.9 cm), and a difference between adult height and predicted height at a value of 0.78 SD (5.2 cm) [12] (see Table 2 below).

When we compared the results of both ISS studies with the male children in our cohort, we found that the total height gain was similar in Group A (7 cm) and in Group B (8 cm) (*p* = 0.181), and was lower in group C (3 cm). No adverse effects were found in any of the treated groups.

IGF1 was lower in patients with GHD (−1.27 SD) in comparison with patients with normal GH levels (0.22 SD), although most deficits diagnosed by stimulation tests did not present an abnormal mean of IGF1. In both treated groups, IGF1 increased with treatment.

Initial sizes were worse in children with lower IGF1 and GHD (−3.17 SD). They also had greater delayed puberty and less pubertal height gain (18.63 cm) than those with normal GH and IGF1 (22.96 cm). There was also a greater overall adult height (−1.33SD) and total height gain in patients with GHD, in addition to lower IGF1 (1.84 SD) (*r* = 0.24) (*p* = 0.034).

Group A were reassessed using stimulation testing with insulin, and 39.4% of those tested were diagnosed with permanent GH deficiencies.

## 4. Discussion

We described two groups of patients who were treated with GH; specifically, one group of subjects who were diagnosed with GHD (Group A), and one group who were diagnosed with ISS (Group B); in addition, there was one sample of children with ISS who were not treated (Group C), which was included to establish the efficacy of the diagnostic tests and of GH treatment in the aforementioned groups.

In our study, Group A and C did not reach their target heights, whereas Group B reached their target heights. It is interesting to note that both treated groups (A and B) revealed normal adult heights, while group C showed lower adult heights. Accordingly, Im et al. stated that both treated groups in their study (GH-deficient patients in the third year of treatment and ISS in the second year of treatment) reached mid-parental height, but at different velocities [16]. However, Kim et al. reported a better response in the GHD group than in the ISS group, without differences in the levels of IGF1 between both groups [17].

The strength of our study is that most patients (90%) had already reached the adult height, thus making it possible to analyze their final evolution.

Although GH has been used to treat short stature due to GH deficiency and other conditions for more than 40 years, the criteria for defining satisfactory responses and targets have never been discovered [18].

Nowadays, GH treatment is approved as a therapy for ISS in the United States, whereas in Europe, it is not approved; thus, it could be interesting to evaluate its utility regarding ISS in a European sample group.

In our study, children with ISS (Group B) showed a GH treatment response that was not different to children with GH deficiencies (Group A). Similarly, some groups revealed that the response to treatment was independent of the etiology of short stature [19,20,21]. The results of the National Cooperative Growth Study (NCGS), comparing 1186 children with treated ISS and 1899 children with treated GH deficiency, likewise showed similar behavior in both groups. Although this may have been due to the effectiveness of the biosynthetic GH, it could also have been caused by the fact that the GH-deficient group contained significant numbers of children with ISS due to problems with the diagnostic tests for GH deficiency [19,22].

On the other hand, Kim et al. reported a better response in the GHD group than in the ISS group, without differences in the levels of IGF1 between both groups [17].

According to the currently published studies on untreated ISS in the European group, in 229 patients, a spontaneous growth was reported at 2 years after previously being 1.7 SD below the mean height based on age, with a total loss of 1 SD during childhood. Furthermore, they revealed a delayed onset of puberty with an acceleration of growth velocity at puberty, reaching a height of −1.5 SD [23,24,25]. Hence, the observed prepuberal and pubertal increases in height in treated patients with ISS, together with the fact that untreated children with ISS showed lower spontaneous height gains, supports our hypothesis that biosynthetic GH increases adult height.

Cochrane’s review established that children with ISS underwent an increase in their final height of 3.7–7.5 cm, but nevertheless, they reached a shorter than average final height [23]. On the contrary, in a meta-analysis of 21 studies, Paltoglou et al. showed that treatment with GH improves short-term linear growth and increases adult height as compared to control subjects without treatment [26]; in our study, the observed adult height was similar to the parental and normal population heights. This supports the hypothesis that different responses occur depending on the type of treatment received.

It is also interesting to propose a classification along the lines of patients with ISS and familial short stature and normal or delayed puberty, and patients with non-familial ISS and normal or delayed puberty. In terms of this classification, European group underwent treatment with GH in all subgroups, and those with familial short stature reached a worse adult height after treatment, but a clear benefit was seen in those without familial short stature who experienced delayed puberty, which suggests a transient deficit of GH that was subsidiary to treatment for a limited time [24,25]. These points also influenced our sample; according to the results of our study, group A showed an earlier onset of puberty than the ISS group due to the patients in group A having constitutional delays in growth and development, while the ISS patients with major differences with GHD adult height had familial short stature.

In this way, Sotos et al. compared 305 untreated ISS controls and 123 children with treated ISS, classified in terms of familial short stature, and constitutional delays in growth and development, and those with normal development and normal family height, and their findings supported the results obtained for the European sample group, with normalization of growth and adult height in all ISS-treated versus untreated children, in addition to a higher growth in those without familial short stature, as in the European sample group [24,25].

Additionally, other publications in favor of treatment of ISS defend a dose-dependent height gain of between 4 and 8 cm, with a mean a gain of 5 cm in men and 6cm in women; this represents an increased growth rate of 4 cm/year as compared to the speed of growth prior to treatment, an increase in height of +2.7 SD as compared to that measured at the beginning of treatment, and a +1.4 SD increase in final height, which indicates that 50.9% of men and 60.8% of women reached parental height or exceeded it after the treatment [1,7,27].

There are also multiple available systematic reviews of controlled and uncontrolled studies conducted in the last 30 years on treated and untreated children with ISS, which reveal dose dependence, with a difference of 1.2 cm in final height in relation to the dosage. In addition, in randomized studies, it was found that treatment increased heights by 4–6 cm, resulting in a mean gain of + 0.78 SD in treated patients and a gain of +0.45 SD (3 cm) in non-randomized groups, in comparison to untreated patients [28]. Nevertheless, it was suggested in the literature that further studies are required for determination of the duration and dose of GH treatment in these children [29]. Dose-dependence data are interesting since this relationship is not statistically significant in the cases of isolated GH deficiency, with this being the case both in our sample and in the studies published so far [2,30,31]. This is a fact that can be explained by the partial insensitivity to GH and IGF1 showed in children with ISS [32].

Rahmati et al. recently published a complete meta-analysis containing cohort-based studies, randomized controlled trials and non-randomized controlled trials regarding growth hormone therapy, in which they concluded that growth hormone therapy can increase the adult height of children with ISS by 3.4–4.2 cm, numbers that are similar to ours. Most of studies were clinical trial studies and the results of these studies are more reliable. Further, there were some studies that had a small sample size; in such studies, the existence of bias is more probable [33].

Finally, it should be noted that in the area of treated idiopathic short stature, the majority are non-randomized studies that involved 2–10 years of monitoring, with samples ranging from 22 to 655 children; these studies were more heterogeneous (both sexes) and had more consistent results than the groups of idiopathic short stature found in our study, although similar results were obtained [16,19,20,21,24,25,28,29]. Instead, our results were influenced by the fact that we added a comparison with GHD, and that we were probably treating ISS patients under the misdiagnosis of GHD.

The limitations of the study are that only the group of patients with GHD was studied prospectively while the comparison groups were historic, with all the limitations that this entails. Apart from its retrospective nature, incomplete data could also be found, in addition to the possibility of inter-observer variability and, perhaps, measurement errors. Additionally, GH deficiency could have been overdiagnosed due to the limitations of current GH testing, bearing in mind that we did not perform sex steroid therapy primarily [7]. Moreover, due to the evolution of laboratory techniques throughout the 14 years of study, there may have been differences in the methodologies of the hormone assays, leading to difficulties in comparing the test results. Accordingly, it is important to consider the introduction of the IS 98/574 standard in 2006, which calibrates mass and reference units for an analyte in chemiluminescent immunometric assays through a combination of monoclonal and polyclonal antibodies with antigens.

Finally, the small number of patients with ISS and the absence of female sex in the sample should also be considered in relation to the comparisons with ISS, both with and without GH treatment.

A practical and theoretical implication of our study concerns the utility of GH in the treatment of ISS, as the results—in comparison with the target height—were similar in both the GH-treated ISS group and the isolated GH-deficiency group (Group A and B). Since GH treatment appears to be both safe and effective, in some cases, we support its use for both GH deficiency and ISS.

Current diagnostic approaches to establish GH deficiency are imperfect, and research should be expanded to identify methods that have greater sensitivity and specificity in order to better conduct diagnoses and establish more appropriate therapeutic indications. Thus, the critical assessment of functional GH testing methods is necessary.

Since patients diagnosed with GH deficiency may not, in fact, be GH-deficient, the conclusions derived from studies of GH-deficient individuals may be inaccurate.

Additionally, despite the great controversy regarding the treatment of both idiopathic forms and those family variants that are considered normal, according to the observed heterogeneous responses to treatment, it would be interesting to extend the experience with ISS to other indications. In this regard, it is interesting that Schena et al.’s study concludes that the selected ISS patients could justify long-term treatment costs [21].

In brief, the adult height observed for subjects with treated isolated GH deficiency is similar to the final height observed in those with treated idiopathic short stature. This could be explained by the possibility that children with ISS were present in the isolated GH deficiency group due to inaccuracies in the currently used diagnostic methods.

## Figures and Tables

**Figure 1 jcm-10-04988-f001:**
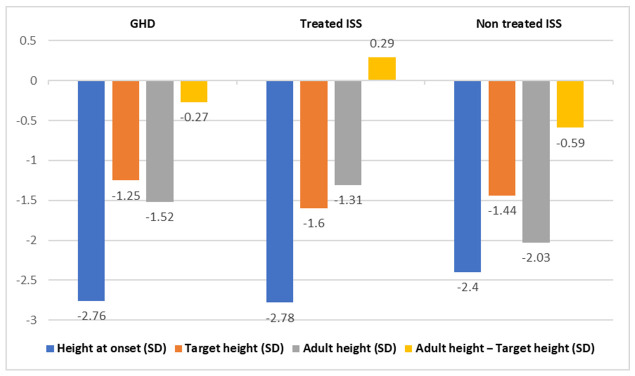
Comparison of variables of GHD patients vs. treated ISS and untreated ISS patients with statistical differences.

**Table 1 jcm-10-04988-t001:** Results and differences between three groups: a group of children with treated Growth hormone deficiency (A), a group of children with idiopathic short stature with spontaneous growth (C), and a group of children with idiopathic short stature treated with GH (B).

	Group A Means (n 67)	Group B [12] Means (n 30)	Group C [11] Means (n 42)	*p* Value
Age at start of GH treatment (years)	9.99	11.1	10.8	0.010 *
Age at onset of puberty (years)	12.5	13.4	13.2	0.000 *
Initial Bone Age/Initial Age	0.71	0.85	0.87	0.217
Height at start of GH treatment (SD)	−2.76	−2.78	−2.4	0.020 *
Target height (SD)	−1.25	−1.6	−1.44	0.050 *
Predicted adult height (SD)	−1.22	−2.09	−2.03	0.240
Height at onset of puberty (SD)	−1.96	−2.38	−2.74	0.090
Pubertal gain (cm)	24.96	26.3	23.75	0.017 *
Adult height (SD)	−1.52	−1.31	−2.03	0.537
Adult height—Target height (SD)	−0.27	+0.29	−0.59	0.001 *
Adult height—Predicted adult height (SD)	−0.3	0.78	0.05	0.596
Adult height—Height at onset of GH treatment (SD) IGF1 (SD) at start of treatment IGF1 (SD) at end of treatment GH posology (mg/kg/day)	1.24 −1.27 0.023 0.028	1.47 0.24 1.63 0.062	0.39 0.22 0.34 -	0.080 0.028 * 0.026 * 0.063

* Statistical significance; NS: no statistical significance; A: treated GH deficiency; B: treated idiopathic short stature; C: untreated idiopathic short stature. The Pearson correlation was used in the correlation of two quantitative variables. Logistic regression was used to investigate the effects of several parameters on dichotomous variables. In addition, for qualitative comparisons with more than two categories and an n of less than 30, the Kruskal–Wallis test was used. On the other hand, for samples with sizes of under 30 and with dichotomous categories, the Mann–Whitney U test was used. Finally, multiple regression was used in comparisons involving adult height.

**Table 2 jcm-10-04988-t002:** Statistical comparisons between children with treated growth hormone deficiency (GHD) (A) and children with idiopathic short stature (ISS) and spontaneous growth (C); between children with treated GHD (A) and a group of ISS-treated children with GH (B); and between children with treated ISS and untreated ISS (B and C).

	*p* Value Group B [12] vs. Group C [11]	*p* Value Group A vs. Group B [12]	*p* Value Group A vs. Group C [11]
Chronological age	0.958 (NS)	0.033 *	0.025 *
Age at onset of puberty (years)	0.816 (NS)	0.022 *	0.017 *
Height at start of GH treatment (H) SD	<0.050 *	0.002 *	0.044 *
Target height (TH) SD	0.903 (NS)	1.797 (NS)	0.045 *
Predicted adult height (PAH) SD	0.324 (NS)	0.290 (NS)	0.274 (NS)
Height at onset of puberty (PH) SD	1.972 (NS)	0.047 *	0.461 (NS)
Pubertal gain (cm)	<0.050 *	0.017 *	0.016 *
Adult height (AH) SD	<0.050 *	0.047 *	0.217 (NS)
AH-H (SD)	<0.050 *	0.181 (NS)	0.415 (NS)
AH-TH (SD)	<0.050 *	0.062 (NS)	1.691 (NS)
AH-PAH (SD)	<0.050 *	0.430 (NS)	0.285 (NS)

* Statistical significance; NS: no statistical significance; A: treated growth hormone deficiency; B: treated idiopathic short stature; C: non-treated idiopathic short stature. Pearson correlation was used in the correlation of two quantitative variables. Logistic regression was used to investigate the effects of several parameters on dichotomous variables. In addition, for qualitative comparisons with more than two categories and an n of less than 30, the Kruskal–Wallis test was used. On the other hand, for samples with sizes under 30 and with dichotomous categories, the Mann–Whitney U test was used. Finally, multiple regression was used in comparisons involving adult height.

## Data Availability

All data are available at the RIUMA thesis repository: https://riuma.uma.es/xmlui/handle/10630/14990 accessed on 26 October 2021.

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
