# Peer review of "Isolated Growth Hormone Deficiency and Idiopathic Short Stature: Comparative Efficiency after Growth Hormone Treatment up to Adult Height"

_jcm, 2021, doi:10.3390/jcm10214988_

Round 1
Reviewer 1 Report
The authors presented important study investigating the issue with qualifying children for growth stimulating treatment with recombinant human GH. In the study they included three independent groups of children, which included children with IGHD treated with GH, children with ISS also treated with GH, and final group of children with ISS without treatment. The study indicated that as in both IGHD and ISS the final heights were similar, yet only the group of GH treated with ISS reached target height while the other two groups did not reach their target heights. Overall the study support and justify considering the cost of providing GH therapy to children with ISS. However, as mentioned by the authors it is difficult to determine the action and regulation of GH in children due to complex physiology, which cause development of differetn stimulatory interventions it would e also of interest to improve the follow during initial treatments to better justify the length and dosage of intervention. The authors mentioned analysis of IGF-1 in treated patients, yet the date was not presented. It would be of interest to present the data if available and if possible to correlate the alterations in IGF-1 with growth rate in each individual patient which could give important knowledge about potential resistance or activation of IGF-1 secretion as an additive for growth rate.
Author Response
Dear Editor, Thank you very much for your suggestions. We attach our corrections to you according to the reviewers' recommendations. We hope they meet your expectations Thank you All the best Ana B. ArizaReviewer 1:
The authors mentioned analysis of IGF-1 in treated patients, yet the date was not presented. It would be of interest to present the data if available and if possible to correlate the alterations in IGF-1 with growth rate in each individual patient which could give important knowledge about potential resistance or activation of IGF-1 secretion as an additive for growth rate.
We have added data of IGF1.
Reviewer 2 Report
The authors present retrospective results showing that as commonly defined by a peak GH <7, having GHD does not predict a greater response to GH in terms of adult height than for idiopathic short stature patients. There are a number of concerns they need to address.
- Title: Since in several places in the paper they refer to their group A patients as having isolated GHD, and it is an important distinction, they should add the word "Isolated" at the beginning
- The introduction spends too much time on discussing the problems inherent in GH stimulation testing which can be shortened. However, a point they should make is that there is disagreement as to whether patients with GHD respond better to GH than patients with ISS, with some of the difference likely due to how strictly GHD is defined. For example, if you look at patients with more than 1 pituitary deficiency or those with severe deficiency (i.e. peak GH < 3 ng/ml) the results are better than when looking at isolated GHD or ISS.
- Page 2 line 62: ambispective is not a word. I think they mean "retrospective"
- On Page 2 line 74 they state "Stimulation tests were performed with exercise plus propranolol, glucagon and clonidine" whereas in line 102 they state "Two GH stimulation tests (exercise or clonidine) with GH < 7 ng/ml or delayed bone maturation of a year or more". This is confusing. Also it appears that if the bone age is delayed by a year or more, they don't need to have a peak GH of <7 to be included in the GHD group, unless they meant to say "GH <7 ng/ml AND a delayed bone age". A bone age delay of >1 year is by itself a very weak indicator of GHD.
- Table 1 should include a comparison of bone age delay for the 3 groups. Also the authors should comment on why the onset of puberty is earlier in patients in group A than the ISS groups, as one would think that for true GHD, the bone age delay would be greater and the onset of puberty would be later. They should also comment on the big difference in PAH for GHD vs ISS, which one would assume is due in large part to the ISS group having many subjects with familial short stature.
- Page 5 line 176: The point is made that the total height gain after GH treatment is similar in groups A and B with a p of 0.181 but that is not clear from table 2. Are they referring to AH-PAH and why notinclude the actual p values rather than just NS. Also in the group A column, all significant differences are given as p<0.05 but not in other columns. Please include the except p values for all entries.
- It would be helpful somewhere to state the differences with GH therapy for groups A and B not just as delta height SD but in cm as well for better comparison with the literature they discuss.
- The authors point out in the discussion that many previous studies they reference have looked at the effect of GH on adult height in patients with ISS and most of those studies include larger numbers of subjects. They need to do a better job of explaining what their study adds to the extensive literature on the subject.
Author Response
Dear Editor, Thank you very much for your suggestions. We attach our corrections according to the reviewers recommendations. We hope they meet your expectations Thank you so much All the best Ana B. Ariza- Title: Since in several places in the paper they refer to their group A patients as having isolated GHD, and it is an important distinction, they should add the word "Isolated" at the beginning
We have added "Isolated" to the title
- The introduction spends too much time on discussing the problems inherent in GH stimulation testing which can be shortened. However, a point they should make is that there is disagreement as to whether patients with GHD respond better to GH than patients with ISS, with some of the difference likely due to how strictly GHD is defined. For example, if you look at patients with more than 1 pituitary deficiency or those with severe deficiency (i.e. peak GH < 3 ng/ml) the results are better than when looking at isolated GHD or ISS.
We have summed up introduction and added that respond to GH could depend to how strictly GHD is defined.
- Page 2 line 62: ambispective is not a word. I think they mean "retrospective"
We show an study with a prospective follow up (GHD group) and two retrospective follow up (ISS groups), so we have detailed that it is bidirectional.
- On Page 2 line 74 they state "Stimulation tests were performed with exercise plus propranolol, glucagon and clonidine" whereas in line 102 they state "Two GH stimulation tests (exercise or clonidine) with GH < 7 ng/ml or delayed bone maturation of a year or more". This is confusing. Also it appears that if the bone age is delayed by a year or more, they don't need to have a peak GH of <7 to be included in the GHD group, unless they meant to say "GH <7 ng/ml AND a delayed bone age". A bone age delay of >1 year is by itself a very weak indicator of GHD.
We have eliminated the mistake of glucagon test and bone age.
- Table 1 should include a comparison of bone age delay for the 3 groups. Also the authors should comment on why the onset of puberty is earlier in patients in group A than the ISS groups, as one would think that for true GHD, the bone age delay would be greater and the onset of puberty would be later. They should also comment on the big difference in PAH for GHD vs ISS, which one would assume is due in large part to the ISS group having many subjects with familial short stature.
We have included the comparison of bone age.
- Page 5 line 176: The point is made that the total height gain after GH treatment is similar in groups A and B with a p of 0.181 but that is not clear from table 2. Are they referring to AH-PAH and why notinclude the actual p values rather than just NS. Also in the group A column, all significant differences are given as p<0.05 but not in other columns. Please include the except p values for all entries.
We have included all p values.
- It would be helpful somewhere to state the differences with GH therapy for groups A and B not just as delta height SD but in cm as well for better comparison with the literature they discuss.
We have added cm.
- The authors point out in the discussion that many previous studies they reference have looked at the effect of GH on adult height in patients with ISS and most of those studies include larger numbers of subjects. They need to do a better job of explaining what their study adds to the extensive literature on the subject.
We have explained what our study adds.
Round 2
Reviewer 2 Report
The authors have taken most but not all of my suggestions and I have a few more that I did not make on the first review.
- Abstract: As requested in my initial review, please replace "ambipective" which is not a word with "retrospective" which I think describes the study well.
- Line 49: same issue
- Line 93: it says 2 stimulation tests but exercise or clonidine implies 1 test. Do they mean exercise AND clonidine or were other tests used?
- Line 138: Please replace 89.58% with 90%. Two decimal places implies a high level of precision- same later in the Ms line 191
- Line 235: familial not familiar short stature
- Line 294: diagnostic APPROACHES
- The reference numbers need to be redone. For example Line 300 Schena et al is reference 21 not 20. Also line 210 Kim et al should be reference 17 not 22 so the authors should recheck all the reference numbers.
- Lines 21-26: In these lines the authors seem to imply that their study will try to provide a more functional test of GH deficiency but they in fact do not. So these comments if included may best be left for the discussion. "Having said that, it would be of interest to use more reliable tests for establishing a better grounded and more trustworthy diagnosis. Since patients diagnosed with GH deficiency may not be in fact GH deficient, the conclusions made from studies of GH deficient individuals may be inaccurate. Thus, the critical assessment of a functional GH testing is necessary."
- Table 1: Should be "initial bone age/chronological age" and I am surprised that group A is not statistically lower than groups B and C. Could the authors recheck that?
- Table 2: They don't need to say NS after all the p values that are >0.05 but I had requested for column 1 (group B vs C) the actual significant p values rather than just <0.05 and the authors have still not done that.
Author Response
Thank your for your recommendations. We attach our changes.
- Abstract: As requested in my initial review, please replace "ambipective" which is not a word with "retrospective" which I think describes the study well.
- We have changed ambispective for retrospective
- Line 49: same issue
- We have changed the issue
- Line 93: it says 2 stimulation tests but exercise or clonidine implies 1 test. Do they mean exercise AND clonidine or were other tests used?
- We have changed or for AND
- Line 138: Please replace 89.58% with 90%. Two decimal places implies a high level of precision- same later in the Ms line 191
- We have changed the percentage
- Line 235: familial not familiar short stature
- We have changed familiar for familial
- Line 294: diagnostic APPROACHES
- We have added approaches
- The reference numbers need to be redone. For example Line 300 Schena et al is reference 21 not 20. Also line 210 Kim et al should be reference 17 not 22 so the authors should recheck all the reference numbers.
- We have redone reference numbers
- Lines 21-26: In these lines the authors seem to imply that their study will try to provide a more functional test of GH deficiency but they in fact do not. So these comments if included may best be left for the discussion. "Having said that, it would be of interest to use more reliable tests for establishing a better grounded and more trustworthy diagnosis. Since patients diagnosed with GH deficiency may not be in fact GH deficient, the conclusions made from studies of GH deficient individuals may be inaccurate. Thus, the critical assessment of a functional GH testing is necessary."
- We have moved this lines to discussion
- Table 1: Should be "initial bone age/chronological age" and I am surprised that group A is not statistically lower than groups B and C. Could the authors recheck that? We have changed chronological age. Results are correct.
- Table 2: They don't need to say NS after all the p values that are >0.05 but I had requested for column 1 (group B vs C) the actual significant p values rather than just <0.05 and the authors have still not done that.
- We have added p values